# Health Equity Implications of the COVID-19 Lockdown and Visitation Strategies in Long-Term Care Homes in Ontario: A Mixed Method Study

**DOI:** 10.3390/ijerph19074275

**Published:** 2022-04-02

**Authors:** Ammar Saad, Olivia Magwood, Joseph Benjamen, Rinila Haridas, Syeda Shanza Hashmi, Vincent Girard, Shahab Sayfi, Ubabuko Unachukwu, Melody Rowhani, Arunika Agarwal, Michelle Fleming, Angelina Filip, Kevin Pottie

**Affiliations:** 1School of Epidemiology and Public Health, Faculty of Medicine, University of Ottawa, Ottawa, ON K1G 5Z3, Canada; ammar.saad@uottawa.ca; 2Interdisciplinary School of Health Sciences, Faculty of Health Sciences, University of Ottawa, Ottawa, ON K1N 6N5, Canada; omagwood@bruyere.org; 3C.T. Lamont Primary Care Research Centre, Bruyère Research Institute, Ottawa, ON K1R 6M1, Canada; rhari077@uottawa.ca (R.H.); ssayf086@uottawa.ca (S.S.); unachukwuubabuko@gmail.com (U.U.); 4Faculty of Medicine, University of Ottawa, Ottawa, ON K1H 8M5, Canada; bjose093@uottawa.ca (J.B.); vgira095@uottawa.ca (V.G.); 5Faculty of Science, University of Ottawa, Ottawa, ON K1N 9B4, Canada; 6Department of Psychiatry, University of Toronto, Toronto, ON M5T 1R8, Canada; syeda.hashmi@mail.utoronto.ca; 7Faculty of Nursing, University of Ottawa, Ottawa, ON K1H 8M5, Canada; mrowh058@uottawa.ca; 8Harvard TH Chan School of Public Health, Boston, MA 02115, USA; arunika@mail.harvard.edu; 9Ontario Centres for Learning, Research and Innovation in Long-Term Care, Bruyère Research Institute, Ottawa, ON K1C 2Z6, Canada; mfleming@bruyere.org (M.F.); afilip@bruyere.org (A.F.); 10Department of Family Medicine, Schulich school of Medicine & Dentistry, Western University, London, ON K1C 2Z6, Canada

**Keywords:** COVID-19, long-term care, visitation strategies, lockdown, health equity, elderly, older adults

## Abstract

The COVID-19 pandemic has negatively impacted the lives and well-being of long-term care home residents. This mixed-method study examined the health equity implications of the COVID-19 lockdown and visitation strategies in long-term care homes in Ontario. We recruited long-term care home residents, their family members and designated caregivers, as well as healthcare workers from 235 homes in Ontario, Canada. We used online surveys and virtual interviews to assess the priority, feasibility, and acceptability of visitation strategies, and to explore the lived experiences of participants under the lockdown and thereafter. A total of *n* = 201 participants completed a survey and a purposive sample of *n* = 15 long-term care home residents and their family members completed an interview. The initial lockdown deteriorated residents’ physical, mental, and cognitive well-being, and disrupted family and community ties. Transitional visitation strategies, such as virtual visits, were criticised for lack of emotional value and limited feasibility. Designated caregiver programs emerged as a prioritised and highly acceptable strategy, one that residents and family members demanded continuous and unconditional access to. Our findings suggest a series of equity implications that highlight a person-centred approach to visitation strategies and promote emotional connection between residents and their loved ones.

## 1. Introduction

COVID-19 has had a distinct and dramatic impact on physical and mental health [1,2], livelihood [3,4], and social relations [5,6], changing the lives of millions of people globally [7]. To curb the spread of the SARS-CoV-2 virus, nations resorted to implementing public health measures, such as enforcing “lockdowns” and recommending physical distancing. While effective, evidence suggests that such measures did not come without negative consequences on the well-being, social life, and mental health of many people; a rapid review early in 2020 highlighted a multitude of psychological stressors experienced by self-isolating individuals [8]. Another scoping review showed a similar negative impact on the well-being of healthcare workers [9]. Among the public, a large-scale cross-sectional survey in the UK showed a high prevalence (35.8%) of loneliness during the COVID-19 pandemic [10]. Social isolation and loneliness among the elderly have been of interest due to the serious impact they have on their well-being, mental and physical health, and longevity [11]. Residents of long-term care homes (hereafter referred to as LTC residents) were particularly vulnerable to the grave consequences of COVID-19, such as hospitalisation and death, because of their age, susceptibility to infections, and congregate living arrangements [12]. Early in the pandemic, the World Health Organization identified LTC residents as having increased vulnerability to social isolation and COVID-19, [13] and recommended taking actions to protect them against the spread of the virus [14]. Data has unveiled a disproportionate transmission of the SARS-CoV-2 virus among LTC residents due to unfair and unjust reasons [10], such as ageing [15] and cognitive impairment [16]. Indeed, LTC residents account for 41% of all COVID-19 related deaths worldwide [17]. In Canada, jurisdiction over health and health care is a shared responsibility between the federal and provincial governments, and LTC homes fall within provincial jurisdiction. The province of Ontario has the largest number of LTC homes in the country, with 626 facilities providing homes, care, and support to more than 115,000 people and their families [18,19]. LTC homes in Ontario were profoundly stricken by the pandemic: The SARS-CoV-2 virus infected 15,455 residents, 7285 staff, and claimed the lives of more than 3985 residents as of August 2021 [20].

In an effort to protect LTC residents from COVID-19, the Ontario Ministry of Health and Long-Term Care restricted visits to LTC homes, preventing most family members and friends from visiting their loved ones [21]. Faced with the unintentional consequences of these ‘lockdown’ measures on residents’ social lives [22], reduced levels of professional care in LTC homes [23], and the backlash from family members and the public [24], decision-makers, both at the government and LTC home level, implemented alternative strategies to traditional in-person visits, such as virtual, window, and outdoor visits across the province, which continued to evolve in response to local public health guidelines [25,26]. In an effort to address LTC residents’ psychosocial and emotional needs, while protecting this vulnerable population from the spread of the virus, “Designated caregiver” programs emerged. These programs recognised a small number of visitors as “essential” caregivers, provided them with public health training on infection prevention and control strategies, and authorised them to access LTC homes and care for their loved ones [27]. While these visitation strategies continued to evolve, little was known about how they were perceived by LTC residents, their family members, and healthcare workers.

Health equity refers to the absence of unnecessary and avoidable health disparities that are unjust and unfair [28,29]. For LTC residents, visits from family members and loved ones constitute a fundamental aspect of their lives that maintains a connection with their community and social network, [30,31] and improves their health and well-being [32]. Restricting or regulating these visits during the pandemic, albeit effective in reducing the spread of COVID-19, may have created or worsened LTC residents’ health inequities compared to their counterparts in the community. There is a need to explore the equity “implications” [33] of rapidly evolving COVID-19 visitation strategies in LTC homes. The lived experience of LTC residents and their caregivers can provide valuable insight and potentially propose solutions to combat future health inequities [34]. Our study emerged to address the urgent need to understand the health equity implications of rapidly evolving COVID-19 visitation strategies in the context of LTC homes in Ontario.

## 2. Materials and Methods

### 2.1. Study Objectives and Research Questions

The objective of this study was to explore the health equity implications of emerging visitation strategies in the context of long-term care (LTC) homes in Ontario during the COVID-19 pandemic. To achieve this objective, we aimed to answer the following research questions:What are stakeholders’ perspectives on the priority, feasibility, and acceptability, as well as implementation considerations (duration, frequency, number of visitors), of different visitation strategies to LTC homes during the COVID-19 pandemic in Ontario?What are the lived experiences of long-term care residents, their family members, and their designated caregivers of the COVID-19 lockdown and visitation strategies in the context of LTC homes in Ontario?

### 2.2. Study Design

We conducted an exploratory sequential (quantitative → qualitative) mixed-methods study design in order to collect participant ratings on the priority, feasibility, acceptability, and implementation considerations of visitation strategies to LTC homes, as well as their lived experiences and visitation stories. Using a pragmatic stance to inquiry, we collected survey and interview data from the same sample and used triangulation to increase the validity of our findings. Further, the qualitative data aimed to complement, contextualise and deepen our understanding of the trends seen in the quantitative component. The methods described herein follow our published protocol [35]. We report our findings according to the Good Reporting of a Mixed Methods Study (GRAMMS) reporting guidelines [36].

### 2.3. Participants and Recruitment

This study took place in Ontario, Canada. We recognised the importance of using an inclusive multi-stakeholder approach to our work [37,38], and thus, identified our target populations as stakeholder groups who are responsible for or affected by visitation strategies to LTC homes in Ontario [37]. Stakeholders included LTC residents, their family members and designated caregivers, as well as clinical and managerial healthcare workers. The term “family members” in the context of our study was inclusive of LTC residents’ “family of choice” and not only biologically/legally related family members. Furthermore, the term “healthcare workers” was inclusive of frontline LTC staff directly providing care to residents (e.g., registered nurses, physicians, personal support workers), and those in managerial and decision-making positions (e.g., executives, directors of care, policymakers). We curated a list of LTC homes serving the population of Ontario [39], and initiated contact with a random sample of *n* = 235 homes from that list for which contact information was available. LTC homes used their internal and external communication channels (e.g., mailing lists, social media platforms) to recruit participants to the study on our behalf. In order to participate in the survey, a participant had to be (a) a resident of the province of Ontario; (b) over the age of majority in the province (i.e., 18 years old); (c) able to communicate in either of the official languages of Canada (i.e., English or French); and (d) fall under the definition of a key stakeholder, as described above. We followed the Total Design Method (TDM) to ensure a higher response rate to our survey, whereby three separate reminders were sent to all those who registered for the survey at 1, 3 and 7 weeks post initial contact [40].

Given that this was an exploratory study, we did not have any “a priori” hypotheses and did not plan for any comparative analyses between stakeholder groups [35]. Rather, we planned to report trends in the data and, therefore, set the survey sample size at *n* = 200 participants. Furthermore, we purposively invited a sample of survey completers to partake in a virtual one-on-one interview. Because the interviews focused on experiences with visitation strategies, we focused only on stakeholders directly engaged in visits (i.e., residents and their family members/designated caregivers). We set the interview sample size at *n* = 15 participants to ensure theoretical data saturation and validity of themes [41,42].

### 2.4. Data Collection

We collected data from participants using online surveys and one-on-one virtual interviews (See Appendix A). The survey was developed by adapting the GRADE FACE instrument [43], a structured approach to assessing stakeholder perspectives on guideline implementations using criteria from the GRADE evidence-to-decision (EtD) framework, such as priority, acceptability, and feasibility [44]. This knowledge translation instrument is constructed as a survey with response options that allow for the collection of quantitative and qualitative data, after guideline release [43]. We chose this instrument as a framework for our survey because it aligned with our mixed-methods design and stakeholder engagement approach to evaluating already-implemented visitation strategies in LTC homes in Ontario. Our survey collected participants’ demographics (e.g., age, gender, stakeholder group, and country of birth) and assessed their ratings regarding the priority, feasibility, and acceptability of six visitation strategies implemented by LTC homes in Ontario at the time of data collection (Table 1). Participants used a 4-point Likert scale to respond to questions with: yes, probably yes, probably no, and no. Further questions explored the implementation preferences of each visitation strategy around the duration and frequency of the visit and the number of visitors allowed. Open boxes allowed participants to share qualitative comments about their responses. 

Semi-structured interviews were grounded in stakeholders’ experiences and used open-ended questions to elicit visitation stories and draw out participants’ perspectives using prompts and follow-up questions (See Appendix A). With participants’ consent, the interviews were recorded and transcribed verbatim using Otter.ai software [45]. Any interviews conducted in French were translated and verified by two team members with a Francophone background.

### 2.5. Data Analysis

#### 2.5.1. Quantitative Data Analysis

We analysed participant demographics using descriptive statistics. To facilitate data interpretation, we (a) analysed quantitative data from the survey for each of the six visitation strategies by dichotomizing categorical responses from the 4-point Likert scale to capture positive (yes and probably yes) and negative (no and probably no) trends; and (b) grouped strategies with the same visitation mode for better narrative fit (see integration and interpretation). We calculated the difference in the percentage of participants who positively rated the priority, feasibility, and acceptability of a certain visitation strategy (or group of strategies) compared to another. Due to the exploratory nature of the study, we did not aim to investigate differences in each of the 4-point Likert scale responses, but we reported within-trend differences (i.e., which scale response was more common under each trend), narratively. Furthermore, open-ended responses to questions about the duration and frequency of visits and the number of visitors allowed were grouped into 3–4 categories, post hoc, based on discussions among team members. All percentage differences were accompanied by a 95% confidence interval [46].

#### 2.5.2. Qualitative Data Analysis

We applied the principles of framework analysis to analyse the qualitative data from interviews and open-ended survey comments [47]. Framework analysis is a five-stage process of familiarisation with the data, identifying a thematic framework, indexing (applying the framework), charting and mapping, and interpretation [48]. We selected the GRADE FACE instrument as our initial coding framework [43], given that the FACE questions formed the basis of the quantitative survey. By using the FACE constructs as the initial deductive coding framework for the qualitative data, we then identified themes that contextualised the quantitative data. This instrument used criteria from the GRADE evidence-to-decision (EtD) framework, such as priority, acceptability, and feasibility, [44] positioning it to serve as a framework to analyse our qualitative interview data. Two team members (O.M., A.S.) applied the framework and then inductively open-coded a random subset of *n* = 5 transcripts, coding anything that might be relevant from as many different perspectives as possible. Codes could refer to substantive things (e.g., particular behaviours, incidents, or structures), values (e.g., those that inform or underpin certain statements, such as a belief in evidence-based medicine or patient choice), and emotions (e.g., sorrow, frustration, love) [47]. After coding the first few transcripts, all researchers involved met to compare the labels they applied and agree on a set of codes to apply to all subsequent transcripts. A total of *n* = 34 codes were grouped together under the FACE categories, which formed the analytical framework. This analytical framework was then applied to all transcripts in duplicate. We used NVivo for all qualitative analyses [49].

#### 2.5.3. Integration and Interpretation

We integrated quantitative and qualitative data at the interpretation stage, allowing for the triangulation of data. All team members participated in the interpretation. Together, the team interpreted the findings through a health equity lens, drawing on their professional and personal experiences within the LTC system. The interpretations were guided by the team’s shared value of person-centered care, which is based on the belief that residents’ views, input, and experiences can help improve overall health outcomes. Although participants were asked about their experiences with each of the six visitation strategies independently (Table 1), our interpretation and comparison of quantitative and qualitative data revealed that participants’ perspectives were determined by the mode of visitation strategy (i.e., in-person visits, such as designated caregivers, outdoor visits, and window visits versus remote visits, such as virtual visits, printed emails, and pre-recorded audio and video messages). We, therefore, elected to integrate and present our results using the aforementioned distinction in the visitation mode (i.e., in-person vs. remote), but also report any discordant findings relating to any of the six visitation strategies independently. Our study integrated the data through a narrative using the weaving approach, which involved writing both qualitative and quantitative findings together on a theme-by-theme or concept-by-concept basis. Further, we developed joint displays, explicitly merging the results from the two data sets through a side-by-side comparison to assess the coherence of the two types of data [50]. This assessment of the fit of integration allowed us to assess confirmatory, inconsistent (outliers), and discordant findings. Finally, we built our concluding equity implication statements guided by the equity statements from the GRADE equity methods series [51].

#### 2.5.4. Researcher Reflexivity

This project was led by a physician (K.P.) who provided care to LTC residents during the COVID-19 pandemic. We collaborated with two members (M.F., A.F.) of the Ontario Centres for Learning, Research and Innovation in Long-Term Care at Bruyère (Ontario CLRI) to promote integrated knowledge translation. To align with the Ontario CLRI’s strategic goal to support innovative and interdisciplinary learning opportunities, we integrated medical and nursing trainees (J.B., SS.H., V.G., M.R.), international medical graduates (A.A., U.U.) and undergraduate students (R.H., S.S.) into our data collection teams. The analysis was co-led by two research associates with quantitative (A.S.) and qualitative (O.M.) expertise.

## 3. Results

### 3.1. Participant Characteristics

A total of *n* = 207 eligible individuals registered to participate in this study between October 2020 and March 2021, of which *n* = 201 completed the survey (completion rate = 97.1%). Participants were at a mean age of 53.5 years (SD 14.03), and the majority identified as female (87.1%) and Canadian-born (83.6%). Four participants (2.0%) identified as LTC residents. Ninety-six participants (47.8%) identified as family members of LTC residents, and an additional ninety-six (47.8%) participants identified as healthcare workers. Five (4.4%) identified as belonging to other stakeholder groups, including members of resident associations and partnerships (*n* = 3); content experts and policymakers (*n* = 2). From survey completers, we purposively selected fifteen participants to complete the qualitative interviews between November 2020 and February 2021, eleven of which were designated caregivers (73.3%), three were family members (20%), and one LTC resident (6.7%). Interview participants showed a mean age of 58.4 years (SD 10.87). Ten participants identified as female (66.7%), and nine were Canadian-born (60%). The characteristics of the participants are presented in Table 2.

### 3.2. Stakeholder Perspectives on Visits to Long-Term Care Homes

We identified four overarching themes from the interviews that we conducted with one LTC resident, 3 family members, and 11 designated caregivers. Qualitative findings were combined with quantitative results from the survey using joint displays and weaved along with the narrative of findings below. Raw data from the survey can be found in Appendix A.

**Theme** **1.**
*The initial restrictions (lockdown) placed on visits to long-term care homes were perceived to be unfair due to the inequitable consequences they had on residents, their family members, and the community at large (Table 3).*


We asked participants about their experiences with visits to LTC homes during the COVID-19 pandemic. The majority of participants reported that the “initial lockdown”, in which visits were restricted, and residents were confined to their rooms, led to many inequities (See Table 3). While participants recognised that the cognitive conditions that residents lived with, such as dementia or Alzheimer’s disease, were inevitable to deteriorate, several reported that the lockdown “*condensed all of these changes*” (Interview O) (See Table 3).

Participants voiced their frustration with the initial lockdown that interrupted communication with residents (See Table 3) and disrupted family ties, causing them to miss important family moments, such as the birth and growing of a new grandchild:

*“… we had our first grandchild … she hasn’t been able to hug them or hold them. And he’s going to be a year and a half … she hasn’t held him in over a year, or he hasn’t sat on her lap in over a year”*.(Interview D)

Once they were allowed to visit, family members described a disturbance in the sense of community within LTC homes. The lockdown and restrictions that followed ceased group activities among residents and prevented visitors from interacting with other residents outside their families. Participants described the “joy” such activities had brought before the pandemic and commented on how “hard” and “difficult” it was to experience this disturbance in the sense of community among residents and their family members (Interview O). One family member described how the relationships within the long-term care community have changed after not being allowed to interact with other residents: 

*“I didn’t only help my mom, I would help everybody else right, but I can’t do that now. So you’d lose some of that relationship, even helping other residents, even talking to other residents, you’re not allowed to. So, you really are there for your resident only, and the relationships with everybody else within the home is slightly different”*.(Interview G)

As such, participants perceived restrictions on visits as unfair and in violation of residents’ right to access their support network. They commented that preventing family members from visiting their loved ones “can never happen again” (Interview B), and called for enhanced legislation (e.g., residents’ bill of rights) to ensure that residents have unconditional access to their support network. Feelings of anger and guilt emerged among participants. One family member highlighted her feelings of guilt, describing the status of her parent as if he’s been “thrown into a cage like an animal” (Interview B). Opposing stances on restricting visits and the inequities that ensued were echoed throughout our interviews and stemmed from family members’ fear that residents’ “time is limited” (Interview O), and commitment towards caring for their loved ones in LTC homes (See Table 3 for quote).

**Theme** **2.**
*Transitional visitation strategies may have alleviated some of the inequitable consequences caused by the initial lockdown, but many were still criticised for their feasibility and limited emotional value (Table 4).*


Participants commented on how certain transitional visitations strategies, such as window and virtual visits, lessened some of the inequities that residents suffered from during the initial lockdown. One LTC resident described that she “*wasn’t depressed as much*” after seeing her family through a window visit (Interview C). Other participants also highlighted that such visits strengthened the connection between LTC residents and their family members:

*“It [virtual visit] helps my parents, it helps us. It also helps us in different ways, not only in showing our love telling them, you know, our love, [but] showing the grandchildren who don’t want to sometimes be there”*.(Interview B)

Many participants highlighted that visitation strategies were highly valued when implemented in a format that allowed for an emotional connection between family members, such as in-person visits (See Table 4). Similarly, when visitation strategies were compared for priority, in-person contact, whether inside or outside the home offered “*benefits of just being close to somebody, even if you’re saying nothing*” (Interview I). Outdoor visits and fresh air were also perceived by family members to be beneficial for the mental health of LTC residents. Survey results on the priority and acceptability of visits showed similar trends (See Table 4). Many participants also described how physical touch played an important role in supporting both the LTC resident and family member’s well-being: “*I wouldn’t be able to live, I wouldn’t be able to breathe, if I didn’t have that [physical] contact with her*” (Interview M). In the absence of physical contact, particularly regarding outdoor and window visits, participants described experiences of tearfulness and mental/emotional distress (See Table 4 for quote).

Every interviewed participant (*n* = 15) commented that the success of visitation strategies was dependent on the LTC residents’ cognitive status and capabilities. Virtual visits, although praised for remedying a severed connection with family members while protecting the health of LTC residents, were perceived as challenging for residents to partake in (See Table 4). This represented a discordance from our quantitative survey results that showed virtual visits as a feasible strategy compared to other strategies (See Table 4) Our participants mentioned several tools which facilitated virtual visits, such as Zoom, FaceTime, and Skype but highlighted that many LTC residents were not familiar with them or able to use them independently. Furthermore, participants highlighted that, for non-verbal LTC residents, in-person visitation strategies were the only meaningful option. In-person visits (indoor and outdoor) also presented some challenges, as the combination of physical distancing and mask-wearing made it difficult for LTC residents to recognise or hear their family members: “*…when we were doing the outside visits … there must have been at least six feet … trying to talk to a senior citizen wearing hearing aids, with us wearing masks. So … [it] was great to see him. Great to connect. But the conversations were difficult*” (Interview O). Nonetheless, family members showed a willingness to follow public health measures and wear personal protective equipment during visits to protect the residents and themselves from the spread of the virus as long as they had clear instructions on these measures: “*if they give us the tools, which is what I’ve told them, if they give us the tools, and the information … then we’re quite happy to follow those rules, we have no problem following those rules*” (Interview A).

Participants frequently offered criticisms regarding the feasibility of certain visitation strategies. One criticism pertained to the visit intake process which they were required to follow in order to visit their loved ones. Participants highlighted that mandated COVID-19 testing, particularly at the beginning of the pandemic, was often challenging due to accessibility issues and delays in receiving results. Participants described waiting up to four hours to complete their test and up to 14 days of delay in receiving their test results. As a consequence, participants described how LTC residents went days without a visit from a family member. Some participants highlighted that these issues were resolved with the introduction of rapid testing. Other participants described complaints regarding the large amount of paperwork required to enter the home on each visit and suggested streamlined electronic check-in options for the future. Finally, one participant reflected that visitors to LTC residents are often older individuals who may also have health conditions and that waiting outside in the heat or cold, without a place to rest was an unacceptable set-up.

Several participants were appreciative of the facilitation services provided by staff, such as nurses, social workers, and personal support workers. These individuals supported resident participation in outdoor, window, and virtual visits. However, many participants emphasised that there simply was not enough staff to facilitate visits with the frequency or length desired by family members and that this role could be task-shifted to family members or essential caregivers: “*My argument with the Director of Care all the time was, well, you need staff, you don’t have enough staff and I’m free. Take advantage of me. I can do this, like anybody, and I want to do it. That’s the difference. And I said it wouldn’t cost you anything for me to come. […] I said you’ve got a caregiver, you’ve got these external resources and you’re blocking access to the facility and to the residents, so you need to open that door and make use of those resources that will take the strain off your staff*” (Interview D). Furthermore, one participant highlighted issues regarding privacy, trust, and ethics when visits were facilitated by an LTC staff member. This participant described the visits as being “*guarded*”, resulting in no trust between the LTC staff and visitors, and consequently no confidentiality, privacy, or intimacy during the visit. They also expressed privacy concerns regarding the discussion of personal business, such as filing annual income taxes or selling a property, whereby “*that kind of stuff that has absolutely no business to anybody else*” (Interview E).

Several participants identified infrastructure issues for window visits and virtual visits. Participants frequently criticised window visits, which often involved poor visibility due to sunshine, cleanliness, or reflections. Window visits were not possible for residents who lived on higher floors, and often the windows themselves did not fully open to allow for conversation between family members. Additionally, a lack of digital infrastructure to support virtual visits led to complaints of dropped calls. The availability of technological devices, such as computers or tablets, was inconsistent; some residents owned their own devices, but for others, there were “*much fewer materials for entertainment […], there’s no television, there’s no phone, there’s no internet*” (Interview K). One participant highlighted that the tablets needed to be shared between residents, citing challenges in scheduling virtual visits.

Many participants highlighted that the cold climate of Ontario, particularly in the northern areas and during the fall and winter, made outdoor and window visits unfeasible year-round. While several participants commented that outdoor visits are great under ideal conditions, many also stated that rain, snow, sun, heat, and insects led to concerns: “*What …[is] the matter with your head, why don’t you go home and get inside? You’re standing out there in that wind and the cold, you’re gonna catch your death of cold*” (Interview E). 

Finally, many participants expressed fears regarding COVID-19 transmission. These fears fell largely into two categories: (i) Many participants were concerned about bringing COVID-19 into the LTC home and increasing their loved one’s risk of contracting the virus. Family members often prioritised wearing clean clothes, avoiding public transportation, and isolating themselves prior to their visits. (ii) Family members were also worried about their own risk of contracting COVID-19 due to their age or pre-existing health conditions. Some family members were not comfortable being exposed to other high-risk individuals or participating in weekly testing, and so chose to abstain from visiting or becoming a designated essential caregiver. While the development and distribution of the vaccine eased some worries, participants expressed concerns regarding their effectiveness against new variants and whether this would impact future opportunities for visits.

**Theme** **3.**
*Designated caregivers emerged as a prioritised strategy to address the health inequity among long-term care residents (Table 5).*


Designated caregivers (essential caregivers) emerged as a fair and important strategy for LTC residents (See Table 5) For LTC residents with dementia and other conditions, essential caregivers were deemed to be important in recognising ill-health or discomfort among their family members in LTC homes, and in stimulating cognition through conversations, which improved not only the cognitive status of LTC residents but also their mental and emotional well-being: “*my sister and I are designated caregivers … And she recognizes us, the majority of time now it [dementia] did improve … once we were allowed in every day, it did improve*” (Interview G). As such, our survey results showed the majority of participants rated designated caregiver visits as a prioritised and acceptable strategy (See Table 5). 

Participants also highlighted that their role as designated caregivers transcended that of a visitor and evolved into advocates for resident rights. One designated caregiver commented on her obligation to advocate better care not only for her mother, but anyone in the LTC home who needed it: “*I don’t just advocate for my mom, I will advocate for anybody who’s in there in terms of the care that I feel that they should be receiving*” (Interview A). Designated caregivers, therefore, emerged as “*one of the partners of care*” (Interview M) that residents require continuous access to, in order to ensure optimal support (See Table 5 for quote).

Of note, when designated caregivers were not permitted to visit, many participants highlighted that virtual visits were critical, as they allowed family members to see and hear one another. A virtual visit, during lockdowns, was described as “*worth its weight in gold*” (Interview F).

**Theme** **4.**
*There is a need for a person-centred approach when determining the duration and frequency of visits and the number of visitors allowed.*


Participants had variable preferences for how long and frequent visits should be and how many visitors should be allowed during a visit (Appendix A). One designated caregiver commented against having limits to the duration of essential visits: “*Why should there be any limitation? because everything you do, helps what is happening there*” (Interview E). Similarly, the majority of participants criticised having set rules for these implementation considerations and highlighted the need for a “person-centred” approach by which the duration and frequency of visits and the number of visitors allowed are adapted to the needs of LTC residents and the circumstances of family members: “*I think it depends on the circumstances. For myself, I [am] absolutely content with once a week, but other people need more than that*” (Interview C).

## 4. Discussion

“Home” transcends our place of living and encompasses a multitude of personal, cultural, and social experiences that shape our identity and influence our psychology [52]. Long-term care facilities are, for all intents and purposes, perceived as “homes” where residents share a community among each other and their family members. The COVID-19 pandemic, lockdown, and transitional visitation strategies dramatically shifted the reality of these homes for LTC residents, disconnected them from the community, and changed their perspectives of LTC homes. For a population predisposed to vulnerability due to their age and weakened physical and cognitive health [12], LTC residents found themselves at grave risk of encountering avoidable health disparities due to their susceptibility to SARS-CoV-2 infection and social loneliness brought by the COVID-19 lockdown [11]. Our study emerged to examine the health inequities that LTC residents experienced during this pandemic and to share their feelings and lived experiences around visitation strategies that shook and redefined their perception of “home”.

Our findings describe a multitude of inequities that LTC residents experienced during the initial “lockdown”, early in the pandemic. Suddenly, visits were forbidden, residents were confined to their rooms, and the resulting isolation deteriorated their mental, physical, and cognitive well-being, and disrupted family and community functions. Similar findings are now emerging from other Canadian provinces, such as Quebec [53], and Alberta [54], and are echoed in Europe [55]. As such, participants described this lockdown as “unfair” and showed emotions of anger and guilt when recalling the state of their loved ones during such stressful times. Transitional visitation strategies, such as virtual visits, ensued, and while they provided a means for interaction between residents and their family members, many participants reported the limited emotional connection that characterised their visits. A mixed-method study investigating “video calling” in nursing homes in the Netherlands highlighted similar trends in perspectives, as participants reported missing physical contact with their loved ones when solely relying on these virtual strategies [56]. Furthermore, many participants complained about window visits and how the windows precluded having meaningful conversations with the residents. This highlights a feasibility issue within LTC homes, as the government of Ontario mandates that windows in LTC homes cannot be opened more than 15 cm [57]. Virtual visits were also criticised for their feasibility issues, as ensuring a successful and meaningful virtual connection with LTC residents required an IT infrastructure that many homes lacked. A mixed-method study among family caregivers of LTC residents living with dementia reported a similar trend, as almost a quarter of responders perceived virtual visits as “ineffective” [55]. Of note, we highlight a discordance between our qualitative finding on the limited feasibility of virtual visits, and our quantitative survey results that show these visits as the most feasible (86.6%), a finding which is also replicated in the published literature [58]. This variance could be attributed to retrospective bias, variability of stakeholder groups responding to the survey and interviews, and differences in IT infrastructure and facilitation capacity among LTC homes needed for a successful implementation of virtual visits. Indeed, some participants spoke of homes providing each resident with their own device to conduct virtual visits, whereas others reported that only a handful of devices were shared among residents. Similarly, while some participants described a dedicated staff member to facilitate virtual calls, others reported that the LTC staff rarely had time to facilitate such visits, especially during outbreaks and emergencies. Furthermore, when comparing visitation formats, in-person visits were perceived to be more valued than remote visits. Our quantitative survey results supported these findings and showed a higher rating for the priority (82.4%) and acceptability (82.4%) of in-person visits compared to remote visits (71.6% and 71.1% for priority and acceptability, respectively).

Family members often play a critical caregiving role in LTC homes [59]. Standing in recognition of this fact, designated caregiver programs emerged as a priority visitation strategy in our findings, allowing family members to access LTC homes and provide care and support to their loved ones. All of our interview participants spoke highly of this strategy, describing the emotional and social value it brought them and their loved ones in LTC homes. Designated caregivers provided LTC residents with personal care and a fair opportunity for emotional support and cognitive stimulation, thus improving their mental, physical, and cognitive well-being. They became care partners, supporting and sharing the responsibilities with LTC staff, and advocating for the rights of LTC residents. Our quantitative survey results solidified these findings, as 90.5% and 90% of participants responded positively to the priority and acceptability of designated caregiver programs, respectively. As such, participants highlighted the importance of providing LTC residents with continuous access to their designated caregivers and emphasised that locking down LTC homes and preventing designated caregivers from fulfilling their role as partners in care cannot happen again. While family caregivers have been historically part of the care of LTC residents [60], they were mostly perceived as “informal” providers of care. The pandemic has shifted this reality as participants echoed that every LTC resident has the right to receive the level of care and emotional support that designated caregivers provide. This requires supporting initiatives that aim to train designated caregivers and equip them with the knowledge and skills needed to support their loved ones. Furthermore, special attention and funding should target culturally and linguistically diverse (CALD) populations, who may require individualised training tailored to their needs and contexts.

In Canada, similar perspectives have spurred the development of a national standard for the co-design and delivery of integrated, resident- and family-centred LTC services across the country [61]. The high level of interest received indicates that this represents an opportunity to shift the conceptualization and delivery of long-term care [62]. Historically, LTC homes have reflected a “visitor” philosophy that treated families as outsiders or dictated the nature of involvement, as opposed to coordinating with family members to meet the needs of all parties [63]. New developments in the national standards of long-term care will adopt a health equity lens and advocate for the health needs of all residents and families [61]. To be successful, however, it is critical to advocate for more attention to be given to equity issues by exposing unfair and unjust harms related to interim COVID-19 visitation strategies [33]. A new multidimensional LTC standard for Canada must focus on LTC construction, care philosophy, culturally and linguistically diverse populations and plan for future pandemics by involving all affected stakeholders, especially residents and their family members, in decision making. Our findings suggest that a thoughtful deployment of strategies to improve LTC residents’ social engagement may mitigate the negative consequences of any future lockdowns or outbreaks, including mental health outcomes.

### The Equity Implications of Visitation Strategies for Long-Term Care Homes

The visitation stories from LTC residents and their family members highlight unfair and preventable health inequities that need action now and in future outbreaks (Table 6):

Our study is unique in that it emerged to address a critical and time-evolving matter that involved a vulnerable population during the COVID-19 pandemic. We engaged multiple stakeholder groups who were directly involved in visitation strategies, such as LTC residents, their family members, and designated caregivers, providing a fair opportunity to share and present their perspectives and lived experiences. These stakeholder groups are the most affected by the COVID-19 lockdown and visitations strategies, and should, therefore, be more meaningfully engaged in the decision-making process about such strategies in the future. Furthermore, we followed a mixed-methods study design to ensure the collection, analysis, and integration of quantitative and qualitative data that truly represented the views of our stakeholders. Our project was sparked by student interest and, as such, was led by a multidisciplinary team of graduate and undergraduate students, supervised by equity experts and healthcare professionals with experience caring for LTC residents. Further, we collaborated with multiple LTC homes, patient and government organisations, and LTC interest groups. Our collaborations will lead the way for future work that improves the equity of visits to LTC residents. Indeed, our committed team of students are now well-prepared to take on future knowledge mobilisation activities in their respective fields, such as designing and implementing a community service-learning program for undergraduate and graduate medical and nursing students with the aim of equipping the newer generation of LTC providers of care with the knowledge and skills needed to ensure LTC residents are cared for appropriately.

This work, however, is not without limitations. Firstly, while our project aimed to address a critical matter, we began this study during one of the most challenging times in the history of long-term care. Our access to LTC homes was limited and the population we targeted was severely impacted, both mentally and cognitively, by the COVID-19 pandemic and the public health restrictions that followed. We believe this was the main reason why only four LTC residents completed the survey and only one was interviewed. Other factors have also, to our belief, hindered our efforts to interview more LTC residents, such as large-scale public inquiries, threats of class action lawsuits, and LTC staff turnover, all of which prevented many LTC homes from collaborating on this project and recruiting their residents to our study. Secondly, because of the public health restrictions implemented in Ontario at the time of undertaking this study, we relied on LTC homes to recruit on our behalf as they saw fit (e.g., internal and external communication with staff, family members, and residents) and thus, we were unable to reliably ascertain the number of individuals reached or to calculate a response rate. We, however, were able to calculate a survey completion rate, taking into account the number of participants who indicated their interest to participate as the denominator. Thirdly, the exploratory and pragmatic nature of our study has allowed us to explore the ever-changing policies around long-term care visitations, but our sample size was limited, and we were not able to analyse between-group differences in survey ratings and qualitative perspectives. Fourthly, our survey and interview were administered in the two official languages of Canada (i.e., English and French), and were not translated into other languages. This may have limited our understanding of the perspectives of culturally and linguistically diverse (CALD) populations and the inequities specific to their context. Fifthly, only two stakeholders with managerial decision-making capacity responded to the survey, thus preventing our findings from covering the perspectives of those with more in-depth insight into the implementation context of visitation strategies. Sixthly, while our findings regarding the impact of the COVID-19 lockdown and other visitation strategies on the physical, mental, and cognitive health of LTC residents resounded across all interviews, they need to be interpreted with caution, as we only relied on self-reported stories and experiences of individuals to describe a decline in these outcomes. Future research should, therefore and when possible, use validated tools and experienced psychometricians to evaluate the true impact of the COVID-19 pandemic on LTC residents’ physical, mental, and cognitive outcomes. Finally, we did not have the resources to conduct a detailed qualitative inquiry with LTC health workers, many of whom lived and suffered the most difficult of events. Our evidence shows that communication between families and LTC homes was disrupted during the lockdown and after, but we do not know the reasons behind this disruption. Future research that focuses on the experiences and stories of health care workers in LTC homes would improve our understanding of this disruption of communication and the factors that further exacerbated it. Despite these limitations, we envision our exploratory work as a first step on the path to addressing the health inequity of LTC residents.

## 5. Conclusions

The COVID-19 pandemic has shone a light on the systemic inequities in long-term care homes, changed our reality, and reshaped our future. Visitation strategies in long-term care homes and designated caregiver programs need to immediately improve to ensure that the health equity of our elderly and loved ones is maintained and protected. While restricting visits may have seemed like the optimal option at the beginning of the pandemic, the magnitude of impact it had on residents, their family members, and the community at large is unfathomable. Designated caregiver programs present a promising innovation to address the health inequities brought by the COVID-19 pandemic on LTC residents, but they must be guaranteed unconditional and continuous access to their essential support network, now and in future pandemics and emergencies.

## Figures and Tables

**Table 1 ijerph-19-04275-t001:** Visitation strategy definitions.

Visitation Strategy	Description
Designated caregivers	An essential visitor designated by the resident and/or their substitute decision-maker to visit and provide direct care to the resident (e.g., supporting feeding, mobility, personal hygiene, cognitive stimulation, communication, meaningful connection, relational continuity, and assistance in decision-making) [27]. Also referred to as: Essential visitors, designated care partners, and essential caregivers.
Outdoor visits	Visitors may visit an LTC resident at an outdoor space/setting, based on scheduling with the homes. Recognising that not all homes have suitable outdoor space, outdoor visits may also take place in the general vicinity of the home [27].
Window visits	Residents can meet a visitor or a small group of visitors at a window within the LTC home.
Virtual visits	Connect by video teleconferencing software, such as Skype, FaceTime or Zoom.
Audio/video recorded messages	Record an audio or video message and send it to an LTC resident for them to watch/listen to.
Printed emails read by staff	Send a letter by email to an LTC resident and an LTC staff reads the letter to the resident.

**Table 2 ijerph-19-04275-t002:** Characteristics of participants.

Characteristic	Survey Participants (*n* = 201)		Interview Participants (*n* = 15)	
	M	SD (Range)	M	SD (Range)
Age	53.51	14.03 (21–85)	58.4	10.87 (29–73)
**Characteristic**	** *n* **	**%**	** *n* **	**%**
**Gender**				
Male	25	12.4	5	33.3
Female	175	87.1	10	66.7
Other	1	0.5	0	0
**Country of Birth**				
Canada	168	83.6	9	60
Other	33	16.4	6	40
**Stakeholder group**				
A family/relative of an LTC home resident	96	47.8	14 *	93.3
Healthcare workers (both clinical and managerial)	96	47.8	0	0
LTC resident	4	2.0	1	6.7
Other	5	2.5	0	0

* of the 14 family members, *n* = 11 identified as designated caregivers.

**Table 3 ijerph-19-04275-t003:** Display of qualitative findings on theme 1: The restrictions placed on visits to long-term care homes were perceived to be unfair due to the inequitable consequences they had on residents, their family members, and the community at large.

This theme was built upon asking participants about their experiences and stories with LTC visits during the COVID-19 pandemic and is seldom qualitative in nature.
Subthemes: The “initial lockdown”
➢ Led to feelings of isolation and loneliness among LTC residents. ➢ Interrupted communication with the homes and prevented family members from obtaining information about residents and the care they received. ➢ Disrupted family ties. ➢ Disrupted the sense of community within LTC homes. ➢ Led to a deterioration of residents’ physical and mental well-being and worsened their cognitive status.
The initial lockdown and restricting visits to LTC residents were perceived as unfair and in violation of residents’ rights to access their support network.
Supportive quotes: *“By the time the isolation was over. I was really feeling down and depressed … I’m very lonesome … And I have noticed other members in other houses in my area here … their confusion has increased. I think that’s because they’ve isolated for so long too.”* (Resident: Interview C) “*… these are our loved ones that are in there, we should be able to go in and… see how they’re doing and to stay in contact with them … all of a sudden, nobody’s coming to visit them anymore… that’s not right. It’s wrong in so many ways. That’s wrong. It’s not fair.*” (Interview A)

**Table 4 ijerph-19-04275-t004:** Joint display of quantitative and qualitative findings on theme 2: Transitional visitation strategies may have alleviated some of the inequitable consequences caused by the initial lockdown, but some were still criticised for their feasibility and limited emotional value. * Participants responded positively with “Yes” or “Probably yes”.

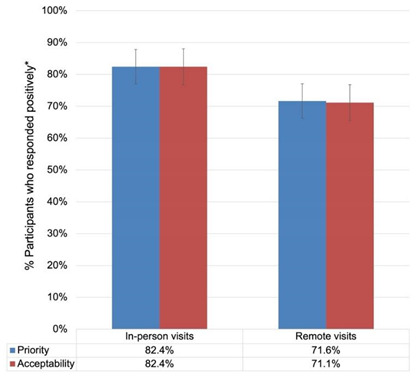	Subthemes:
	➢ Transitional visitations strategies have lessened some of the inequities that residents suffered from during the initial lockdown and strengthened the connection between LTC residents and their family members. ➢ Visitation strategies were highly valued when they allowed for emotional connection. In-person interactions, such as designated caregiver, outdoor, and window visits were prioritised and perceived to be more valuable than remote interactions, such as virtual visits, pre-recorded audio and video messages, and printed emails.
Comparison 1: In-person visits were rated to be more prioritised than remote visits: Difference = 10.8%; 95% CI [2.64%, 18.96%]. Within-trend differences: More “yes” than “probably yes” responses; more “probably no” than “no” responses. (See Appendix A) Comparison 2: In-person visits were rated to be more acceptable than remote visits: Difference = 11.3; 95% CI [3.12%, 19.48%]. Within-trend differences: More “yes” than “probably yes” responses; more “no” than “probably no” responses for in-person-visits, and the opposite for remote visits. (See Appendix A)	Supportive quote: *“We couldn’t hug her. We couldn’t touch her…, you know, hold hands or anything. And we had to stay six feet apart. And yes, she was there, but without that, that social contact. It did affect … affect her … both physically and emotionally”*. (Interview H)
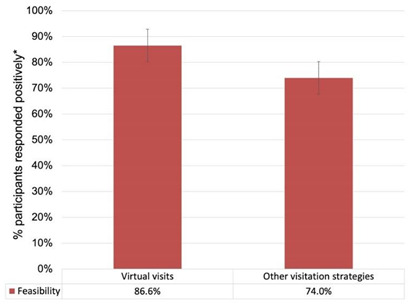	Subthemes:
	➢ The success of visitation strategies was influenced by LTC residents’ cognitive status and capabilities. ➢ Participants frequently offered criticisms regarding the feasibility of certain visitation strategies, such as the visit intake process, not enough staff to facilitate visits, infrastructure issues, the cold climate of Ontario, and fears regarding COVID-19 transmission. ➢ Discordance: Many participants criticised the feasibility of virtual visits and described how conditions, such as dementia, physical disability, vision impairment, and hearing loss made it challenging for LTC residents to fully participate in virtual visits.
Comparison 3: Virtual visits were rated to be more feasible than other visitation strategies: Difference = 12.6%; 95% CI [4.92%, 20.28%]. Within-trend differences: More “yes” than “probably yes” responses; more “probably no” than “no” responses. (See Appendix A)	Supportive quote: *“they [virtual visits] were not successful for her because of dementia … she wasn’t used to seeing somebody virtually … my mom, just was not familiar with the technology. So she wouldn’t really. She’d be talking to somebody on the screen but she wouldn’t know who it was, or she just was not used to that”*. (Interview L)

**Table 5 ijerph-19-04275-t005:** Joint display of quantitative and qualitative findings on theme 3: Designated caregivers emerged as a prioritised strategy to address the health inequity among long-term care residents. * Participants responded positively with “Yes” or “Probably yes”.

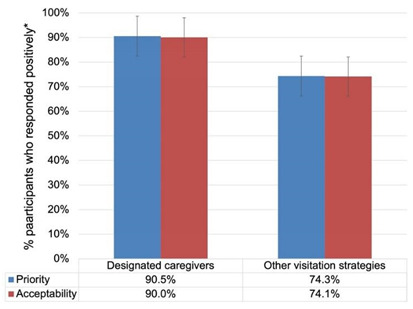	Subthemes:
	➢ Designated caregivers (essential caregivers) provided LTC residents with fair access to their support network and addressed inequities related to social isolation and physical and mental deterioration. ➢ Family members’ role as designated caregivers transcended that of visitors and evolved into advocates for resident rights and partners in their care circle.
Comparison 4: Designated caregivers were rated to be more prioritised than other visitation strategies: Difference = 16.2%; 95% CI [8.93%, 23.47%]. Within-trend differences: More “yes” than “probably yes” responses; more “probably no” than “no” responses. (See Appendix A) Comparison 5: Designated caregivers were rated to be more acceptable than other visitation strategies: Difference = 15.9%; 95% CI [8.56%, 23.24%]. Within-trend differences: More “yes” than “probably yes” responses; more “probably no” than “no” responses. (See Appendix A)	Supportive quote: *“…as essential caregivers, designated caregivers, the residents always need to have access to us … they need access to their support system. You just can’t take all their support that they’ve had over the years … So, when you take it away, it really affects them, and being able now as an essential caregiver to go in, I recognise more and more that this can never happen again”*. (Interview G)

**Table 6 ijerph-19-04275-t006:** Health equity implications of visitation strategies for long-term care homes (with justification from the evidence).

Health Equity Implications	Justification
Participants highlighted the unfairness and negative unintended consequences of the initial lockdown, and emphasised that such aggressive measures cannot be implemented again.	Locking down homes disconnected LTC residents from their family members, friends, and community and confined them to their rooms, often without compensating care. This lockdown sparked feelings of loneliness and isolation and had a detrimental effect on their well-being.
Future visitation strategies should be designed to maintain emotional value for LTC residents and their family members, allowing for in-person interactions in a safe and visit-friendly environment.	Transitional visitation strategies may have provided means to connect LTC residents to their family members, but participants highlighted that they lacked the emotional value needed to sustain their benefit for both the resident and care partners in the long run.
Designated caregiver programs may provide LTC residents with emotional connection and family caregiving, but such programs must also be accessible and adapted to the needs level and context of residents and their family members.	Participants emphasised that many LTC residents need sustained caregiving from their support network. They highlighted designated caregivers as the most prioritised and most acceptable form of visitation.

## Data Availability

Raw aggregate data is presented in Appendix A. To protect the confidentiality of participants, we will not share individualized data beyond what is presented in this manuscript.

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
