# Peer review of "Health Equity Implications of the COVID-19 Lockdown and Visitation Strategies in Long-Term Care Homes in Ontario: A Mixed Method Study"

_ijerph, 2022, doi:10.3390/ijerph19074275_

Round 1

Reviewer 1 Report

Minor Comments:

  1. The authors collected surveyed data using a 4-point scale, but then state they dichotomized the scale responses for ease of interpretation. If this was the case then they should have only collected data as yes/no and so this reads as an ad-hoc modification that was not originally intended. My concern is that the results would look different using a 4-point scale rather than 2-point artificially dichotomized scale. How do the results change?

Major Comments:

  1. The total N was 201 from 207, but how many did not respond to the outreach efforts initially? This is the true response rate where the 97.1% rate is the survey completion rate.
  2. There were very few stakeholders (n = 2) in the decision-making capacity regarding visitation strategies (e.g., mangers, CEOs) and so the paper is slightly overselling the weight of having multiple different stakeholders. It appears they have key data on people, residents and staff, who experienced these strategies but not those who have the ability to speak on their justification for feasibility, acceptability and implementation of these approaches. There should be mentioning of this as a re-framing when possible to convey this point.

Author Response

Thank you for reviewing our work. Please see the attachment for our detailed response.

Reviewer 2 Report

Long-term care (LTC) homes are more than just nursing facilities, they are a veritable 'home' for residents, with a shared community of the residents, their families and care workers. This study is to be commended for exploring the impact of the COVID-19 pandemic through a comprehensive survey and subsequent analysis, discussing the relative risk of infection/morbidity to that of lowered mental/social health conditions, and providing implications for the wellness of those in need of care. Based on the understanding, the following points are taken into account and further improvements are expected:

Although the study is defined as exploratory, previous research needs to be reviewed more carefully. It would be helpful to investigate the pandemic caused by COVID-19 in the context of social health and wellness management, not limited to the scope of LTC, to reinforce the argumentation process of this study.

The utility of virtual visits is a core finding of this study. While the analysis is currently qualitative and descriptive, it should be further explored and compared quantitatively with the benefits of face-to-face visiting strategies. The advantages of virtual visits over face-to-face visits should also be mentioned (e.g. Elimination of travel difficulties, reduction of infection risk). Furthermore, it is advisable to describe what tools were used during the virtual visit and how they were used, as far as records are available.

It is also recommended to discuss in depth, as far as possible, the changes and challenges in communication between the resident and his/her family and the care staff.

Author Response

Thank you for reviewing our work. Please see the attachment for our detailed response

Reviewer 3 Report

Comments and suggestions for authors

The topic presented in the manuscript entitled ‘Health equity implications of the COVID-19 lockdown and visitation strategies in long-term care homes in Ontario: A mixed method study’ is very interesting. The authors present how the changes in visitation strategies (generated by isolation measures in the COVID 19 pandemic lockdown) had a negative impact on the physical and psychosocial well-being of long-term care home residents and their families. The research conducted by the authors was not simple, especially since it took place during the pandemic. I appreciate the work of the authors and the fact that they have shown various aspects related to health inequities and especially that they are already outlining possible directions for action to solve them.

Introduction

It is well written and supported by the research literature and clearly presents the factual situation in the country and province referred to in the study

Materials and methods

The study’s aims, design, the procedure of participants’ recruitment are obviously described. Still, the authors should add some information related to the GRADE  FACE instrument (briefly describe what it entails).

Table 1 is a clear presentation and description of visitation strategies.

The authors used a mixed method which is sufficiently described in Data analysis.  

Results

The authors should state more clearly in the manuscript text whether the 15 interviewed participants are from outside the group of 201 survey participants.

Even if it is a qualitative study, only one participant is a long-term care (LTC) resident, most of the others being designated caregivers and family members. In this case, the authors should mention why there is only one LTC resident. Also the sentence from lines 240-241 should be slightly changed accordingly.  

It stands to reason that the restriction of family member’s visits and the lack of interactions with other LTC residents led to a significant negative emotional, social and physical impact on participants. However, the authors should mention that, at least in terms of cognitive and physical decline, the study takes into account subjective aspects and perceptions of others, and that these are not documented by medical reports or objectively evaluated.

The ‘Boxes’ are appropriate in the manuscript, are clear, concise, present the important aspects of the qualitative analysis, interviews as well as the quantitative analysis. They are a great way to highlight the key issues identified by this research.  The authors should remove from the contents of the manuscript or present more briefly those texts and statements which are found in the ‘Boxes’. Some of them are found in identical form in the text of the manuscript and in the ‘Boxes’, which may bore some readers.

I appreciate the systematic presentation of the themes resulted in the present research.

Discussions

They are well and clearly presented. Though, I have a remark to make:  COVID 19 pandemic, lockdown, different restrictions had and still have a negative impact on all or most people in all aspects of their lives. And this is already mentioned in the beginning of Introduction and Discussions chapters. I suggest that the authors should highlight why this negative impact may be increased for LTC residents (e.g. to detail about the vulnerable population, lines 565-566).

I appreciate that the authors involved students in their research and that in this way they can be in contact with both theoretical and practical aspects related to health equity.

Author Response

(The authors gave the same response as above.)
